# Ten-Year Results of Accelerated Partial-Breast Irradiation with Interstitial Multicatheter Brachytherapy after Breast-Conserving Surgery for Low-Risk Early Breast Cancer

**DOI:** 10.3390/cancers16061138

**Published:** 2024-03-13

**Authors:** Nieves G. Rodríguez-Ibarria, Beatriz Pinar, Laura García, Auxiliadora Cabezón, Dolores Rey-Baltar, Juan Ignacio Rodríguez-Melcón, Marta Lloret, Pedro C. Lara

**Affiliations:** 1Radiation Oncology Department, Dr. Negrin University Hospital Las Palmas GC, 35010 Las Palmas de Gran Canaria, Spain; ibarria.nieves@gmail.com (N.G.R.-I.); beapinsede@hotmail.com (B.P.); laura_garcab@hotmail.com (L.G.); sulipons@hotmail.com (A.C.); lolabaltar@gmail.com (D.R.-B.); nachorodriguezmelcon@hotmail.com (J.I.R.-M.); 2Medical School, Las Palmas University, 35001 Las Palmas de Gran Canaria, Spain; 3Oncology Department, Canarian Comprehensive Cancer Center, Fernando Pessoa Canarias University, 35001 Las Palmas de Gran Canaria, Spain; 4Canarian Insitute for Cancer Research, 380204 San Cristobal de La Laguna, Spain

**Keywords:** partial-breast irradiation, brachytherapy, early breast cancer, breast cancer

## Abstract

**Simple Summary:**

Accelerated partial-breast irradiation (APBI) has gained acceptance in the last few years as a postoperative treatment after breast-conserving therapy. Between December 2008 and December 2017, 182 low-risk breast cancer patients treated by BCS and APBI using interstitial multicatheter brachytherapy were included in this study. After a mean follow-up for survivors of 10 years, the treatment was shown to be safe, as no severe acute/late toxicity (grade ≥ 3) was observed. The 10-year ipsilateral breast tumor recurrence (IBTR) was 1.7% (95%CI: 0.7–2.7%), and the cause-specific survival was 94.9% (95%CI: 93.2–96.6%). We suggest that multicatheter brachytherapy after BCS is safe and effective in early breast cancer patients.

**Abstract:**

Patients with an early carcinoma of the breast are commonly treated by breast-conserving surgery (BCS) and postoperative radiotherapy. Partial-breast irradiation has gained acceptance in the last few years. Between December 2008 and December 2017, 182 low-risk breast cancer patients treated by BCS in the four university hospitals of the province of Las Palmas and treated with APBI using interstitial multicatheter brachytherapy were included in this study. After a mean follow-up for survivors of 10 years, the treatment was shown to be safe, as no severe acute/late toxicity (grade ≥ 3) was observed. The 10-year IBTR was 1.7% (95%CI: 0.7–2.7%), and the cause-specific survival was 94.9% (95%CI: 93.2–96.6%). We suggest that multicatheter brachytherapy after BCS is safe and effective in early breast cancer patients.

## 1. Introduction

Postoperative Whole-Breast Radiation Therapy (WBRT) is considered the standard treatment after Breast-Conserving Surgery [1]. Treatment protocols increasing the dose per fraction and reducing the total treatment time (hypofractionated and ultrahypofractionated radiotherapy) demonstrated similar results to conventionally fractionated radiotherapy [2,3,4,5].

In recent years, Accelerated Partial-Breast Irradiation (APBI) has been gaining acceptance, as this treatment approach would reduce the breast volume irradiated, the treatment time, and the unneeded radiation exposure to healthy tissues like the heart and the lung [6].

Results from phase II [7,8,9] and phase III trials [10,11] studying interstitial multicatheter APBI have shown that the observed clinical outcomes are similar to those achieved by whole-breast irradiation (WBRT). These results support the recommendations from the European Society of Radiotherapy and Oncology/Groupe Européen de Curiethérapie (GEC–ESTRO) [12], the American Society of Therapeutic Radiology and Oncology (AS–TRO) [13], and American Brachytherapy Society (ABS) [14] to select candidate patients for this APBI approach.

The largest published phase III trial from GEC–ESTRO [11] showed non-inferior 10-year figures of ipsilateral breast tumor recurrence and survival rates when comparing APBI to whole-breast irradiation. The cumulative incidence local recurrences at 10 years was shown not to be inferior in the APBI group at 3.51% (95%CI: 1.99 to 5.03) compared to 1.58% (95%CI: 0.37 to 2.78) in the whole-breast irradiation group. These results strongly support the efficacy of APBI in this particular clinical situation. Predefined clinical outcomes also included regional recurrences. Again, no differences were found in the cumulative incidence of regional (lymph node) metastasis at 10 years in the whole-breast irradiation group at 0.39% (95%CI: 0.00 to 0.94) versus the 1.19% (95%CI: 0.31 to 2.06) observed in the APBI group (*p* = 0.15). The distant metastases’ incidence after a 10-year follow-up was statistically similar in both treatment arms at 2.17% (95%CI: 0.90 to 3.44) in the whole-breast irradiation group vs. 2.60% (1.30 to 3.90) in the APBI group (*p* = 0.72). The 10-year disease-free survival was 87·95% (95%CI: 85.10 to 90.91) with whole-breast irradiation and 84.89% (81.97–87.91) with APBI (difference: –3.06%; 95%CI: 7.22 to 1.09; *p* = 0.18).

Data on severe (grade ≥ 3) side-effects after a 10-year follow-up have also been described [11]. The most common type of grade 3 adverse event in both treatment groups was fibrosis (2% for the whole-breast irradiation group vs. 1% for the APBI group, *p* = 0.56). No grade 4 or grade 5 toxicities were observed. Interestingly, patients in the APBI group showed a lower incidence of treatment-related late side-effects when grade ≥ 2 late toxicities were analyzed (*p* = 0.021) [11].

Data on cosmetic results after a 5-year follow-up are available [15]. A total of 413 out of 454 patients (91%) in the whole-breast irradiation group were considered to have excellent to good cosmetic results versus 498out of 541 (92%) patients in the APBI group (*p* = 0.62). Cosmetic results were also scored by the treating physicians. A total of 408 out of 454 patients (90%) in the whole-breast irradiation group and 503out of 542 patients (93%) in the APBI group were scored as showing excellent to good cosmetic results (*p* = 0.12).

Patient-Reported Outcomes (PROs) are also an important study endpoint when comparing APBI versus WBRT. PROs were evaluated in the GEC–ESTRO trial [16] by two quality-of-life (QoL) questionaries: the European Organization for Research and Treatment of Cancer’s (EORTC) QLQ-C30 and the breast cancer module’s QLQ-BR23. These questionaries were completed before radiotherapy, immediately after radiotherapy, and during follow-ups. Major results observed after a 5-year follow-up show that the global health status determinant of QoL was similar at diagnosis and after a 5-year of follow-up in both groups. No differences were found in any other of the QoL scale determinants, except for the breast symptoms scale. Breast symptoms were scored by the patients to be significantly worse after whole-breast irradiation when compared to those observed after APBI, either at the end of the radiotherapy treatment (*p* < 0.0001) or after a 3-month follow-up (*p* < 0.0001), respectively [16].

The available evidence described above support that, at present time, APBI using multicatheter brachytherapy after BCS in patients with early breast cancer is a valuable alternative to WBRT in terms of treatment efficacy and is associated with excellent cosmetic results, better quality-of-life scores, and fewer late side-effects.

The present study was aimed to assess the long-term results at a median follow-up of 10 years, including ipsilateral breast recurrences and survival rates, of patients with low-risk breast carcinomas after breast-conserving treatments.

## 2. Patients and Methods

Patients suitable for a breast-conserving surgery, which was performed in the four University Hospitals of the province of Las Palmas (the University General Hospital of Fuerteventura, the University General Hospital of Lanzarote, the University Materno-Insular Hospital of Gran Canaria, and the University General Hospital of Gran Canaria Dr Negrín), were included in this prospective study.

The eligibility and exclusion criteria have already been published [17]. In short, women aged ≥50 years who were affected by early (pT ≤ 3 cm, pN0, M0) breast cancer and who had undergone BCS, an axillary dissection, or a sentinel node biopsy with microscopically clear resection margins were included. Patients with a lymphovascular invasion were also allowed in the trial. Pregnant or lactating patients, those previously affected by breast cancer or other malignant diseases, and those with extensive intraductal in situ carcinomas, multifocal breast cancer disease, or who were diagnosed with Paget disease were excluded from this study.

The interstitial multicatheter brachytherapy treatment was performed at the Radiation Oncology Department of the University Hospital of Dr Negrín. Patients were prospectively treated under the Spanish RD1566/1998 regulation for Radiation Therapy Quality Assurance. This study was approved by an ethics committee, and informed consent was obtained from all patients.

### 2.1. Procedures

The treatment protocol has been previously reported [17]. In short, all patients had interstitial multiplanar implants to effectively cover the clinical target volume (CTV), including both the surgical cavity and a safety margin (at least 20 mm). All patients had pre-planning computed tomography (CT) and planning CT scans (for the treatment planning and documentation of multicatheter brachytherapy). A standard 1.5 cm catheter spacing was used. Dose prescriptions and calculations were in agreement with report 58 of the International Commission of Radiation Units and Measurements (ICRUs).

A dose–volume histogram analysis was performed to evaluate dose coverage (100% of the dose covered at least 90% of CTV), homogeneity (non-uniformity ratio below 0.35), and normal tissue limits (skin dose below 70% of the prescribed dose). Accelerated partial-breast irradiation was delivered with high-dose-rate (HDR) multicatheter brachytherapy in 8 fractions of 4 Gy, which were always separated at least 6 h from each other. The treatment was administered in 5 consecutive days to a total dose of 32 Gy.

A follow-up [17] was performed jointly by the surgeons and medical oncologist from the participating institutions and the treating radiation oncologists. This clinical examination included the documentation of late adverse effects using the CTCAE 4.0 scale.

An adjuvant systemic treatment was prescribed according to local treatment protocols in every referring hospital, following multidisciplinary-team and international-guideline recommendations.

### 2.2. Outcomes

Long-term ipsilateral IBTR was the primary objective of the present study. The secondary endpoints were (a) incidences of regional recurrence, (b) incidences of distant metastasis, and (c) survival (cause-specific survival and overall survival).

The Kaplan–Meier method was used for constructing survival curves. A statistical comparison was performed by using the log-rank test. A probability level of 0.05 was considered statistically significant. Statistical analyses were performed using SPSS version 26 (IBM Corp., Armonk, NY, USA).

## 3. Results

Between 1 December 2008 and 11 December 2017, 182 women with early-stage breast cancer after BCS were included in the present study and fully completed the protocol for APBI using multicatheter brachytherapy [17].

The patients’ characteristics have been previously described [17] (Table 1). The mean age at treatment was 67 years (range 50–92), and most patients showed a luminal molecular subtype cancer (95.6%). Most patients were considered cautionary or ineligible by ASTRO guidelines but low risk for GEC–ESTRO characteristics.

The follow-up ended on 15 October 2023. The mean follow-up for survivors was 123.27 ± 29 months (median: 123 months; range: 17–177). Four patients were excluded from the follow-up at 17, 42, 46, and 51 months after treatment, as they moved on to another Spanish region.

Three out of the 182 patients developed a local recurrence (LR) during the follow-up, with two relapses being in the tumor bed (23 and 74 months) and the other elsewhere in the ipsilateral breast at 14 months. The IBTR risk at 5 years was 1.1% (95%CI: 0.3 to 1.9) and 1.7% (95%CI: 0.7 to 2.7%) at 10 years.

One case of an isolated regional recurrence was reported, with a 5- and 10-year cumulative incidence rate of 0.6% (95%CI: 0.0 to 1.2%). Two other cases showed axillary relapses with a concomitant systemic relapse at 76 and 153 months, respectively. Five patients showed distant relapses at 14, 22, 24, 76, and 153 months, respectively. The cumulative incidence of distant metastases at 5 years and 10 years were 1.7% (95%CI: 0.8 to 2.6) and 2.3% (95%CI: 1.2 to 3.4), respectively (Table 2).

During the follow-up, three patients developed contralateral breast cancer (1.65%) and thirteen patients developed second primary cancer (7.14%). The 5- and 10-year disease-free survival rates were 96.7% (95%CI: 95.4 to 98) and 94.9 (95%CI: 93.2 to 96.6), respectively. Thirty-six patients died during the follow-up, four of them due to breast cancer and 32 patients by other non-breast cancer diseases. The 5- and 10-year cumulative cause-specific survival rates were 98.3% (95%CI: 97.3 to 99.3) and 97.7% (95%CI: 96.6 to 98.8), respectively (Table 2). The overall survival at 5 and 10 years were 93.3 ± 1.9% and 80.7 ± 3.2%, respectively.

Patients suffering from tumors showing a lymphovascular invasion had a 10-year cumulative IBTR rate of 16% (95%CI: 5.4 to 26.6) compared to the 0,6% (95%CI: 0 to 1.2) of those cases without a lymphovascular invasion (*p* < 0.0001). Tumors showing a high proliferating status (Ki67 positive) showed a 10-year cumulative IBTR of 11.1% (95%CI: 5.0 to 17.2%) vs. the no IBTR of tumors with a low proliferation (Ki67 negative) (*p* = 0.001). None of the low and intermediate ESTRO risk-category patients suffered a local relapse compared to 7.3% (95%CI: 3.2 to 11.4) of those classified as high ESTRO risk (*p* = 0.007). Finally, patients showing a negative estrogen receptor status showed a 10-year IBTR of 10% (95%CI: 0.5 to 19.5) versus the 1.2% (95%CI: 0.3 to 2.1%) of those with a positive estrogen receptor (*p* = 0.028) (Table 1, Figure 1).

The treatment-related acute and late toxicity rates have already been reported [17]. After a mean follow-up of more than 10 years, no severe grade 3–4 late adverse effects were recorded.

## 4. Discussion

Postoperative radiation therapy after BCS reduces the risk of ipsilateral breast recurrences and increases breast cancer survival [1]. Shortening the treatment time, reducing the normal tissue volume irradiated, and improving quality of life are major challenges already accomplished by multicatheter interstitial brachytherapy APBI [10,11,15,16]. This de-scalation in treated volumes should be strictly restricted to appropriately selected patients following the guideline recommendations of ESTRO [12], ASTRO [13], and ABS [14].

Here, we present the updated long-term follow-up (10-year) results of our multicenter study. The local control observed in our follow-up series confirms our preliminary results of an IBTR of 1.1% at 5 years [17] and demonstrated an excellent 1.7% (95%CI: 0.7 to 2.7%) IBTR rate at the 10-year follow-up. Our data compare favorably with the mature data of the APBI arm from the GEC–ESTRO trial, which showed a 3.51% (95%CI: 1.99 to 5.03) IBTR rate at 10 years [11]. In fact, after a similar mean follow-up (10.3 years), our results are closer to the WBRT arm of the trial (IBTR: 1.58%; 95%CI: 0.37 to 2.78). We suggest that our excellent results may also be influenced by the patient characteristics in our series. In fact, our patients showed slightly better characteristics in terms of prognosis than those included in the GEC–ESTRO trial. Patients were older in our study: the mean age was 67 years (94% were postmenopausal) in our series compared with 62 years (83% were postmenopausal) in the GEC–ESTRO trial. The patients showed less lobular and in situ carcinomas in our series (15.9%) compared with 19% in the GEC–ESTRO trial. Finally, no node-positive patients were included in our series of patients, compared to the 1% node-positive patients and the 5% of cases with no lymph node dissection (Nx) in the GEC–ESTRO trial. On the contrary, our series included patients with free surgical margins of less than 1 mm (15 patients out of 182, 8.2%) and a significant number of cases with lymphovascular invasions (15 patients out of 182, 8.2%) (Table 1).

The results of our treatment protocol, with respect to regional relapse (0.6%) and contralateral breast relapse (1.65%), are in good accordance with the GEC–ESTRO results [11]. Our cumulative incidence of regional relapse of 0.6% (95%CI: 0.0 to 1.2%) falls off between the observed cumulative incidence of regional (lymph node) metastasis at 10 years of 0.39% (95%CI: 0.00 to 0.94) in the whole-breast irradiation group and the 1.19% (95%CI: 0.31 to 2.06) in the APBI group of the GEC–ESTRO study. Again, the cumulative incidence of distant metastasis at 10 years in our series of patients was a 2.3% (95%CI: 1.2–3.4) fall off between the observed cumulative incidence of distant metastases of 2.17% (95%CI: 0.90 to 3.44) in the whole-breast irradiation group and the 2.60% (1.30 to 3.90) in the APBI group. Our 10-year disease-free survival was 94.9% (95%CI: 93.2–96.6), which is slightly better than the 87.95% (95%CI 85.10 to 90.91) observed in the whole-breast irradiation group and the 84.89% (81.97–87.91) in the APBI group. The favorable characteristics of our series of patients, especially with reference to the nodal status, support these excellent results.

Interestingly, the overall survival is lower in our series 80.7% (95%CI: 77.5 to 93.9) compared to those observed in the reference phase III trial [11]. In the GEC–ESTRO trial, the 10-year overall survival was 89.52% (95%CI: 86.87 to 92.25) in the whole-breast irradiation-treated patients and 90.47% (88.09 to 92.91) in the APBI-treated patients. The advanced age at diagnosis of the included patients (37.9% of the patients were older than 70 years at diagnosis) would explain these lower survival rates in the case of excellent cancer control figures.

An analysis of the predicting factors of IBTR was performed in our mature follow-up group of patients. The lymphovascular invasions (*p* < 0.0001) and high-proliferation rate, as estimated by Ki67 (*p* = 0.001), resulted in major predictors of a local relapse. This would be anticipated according to proposed guidelines concerning lymphovascular invasions [12], but no data are available regarding the role of tumor proliferation predicting relapses after a multicathether brachytherapy APBI. Only one report from intraoperative radiotherapy with electrons demonstrated a predictive value for local relapses for the Ki67 tumor-proliferation index [18].

Furthermore, all three IBTRs were observed in the ESTRO high-risk group vs. none in the low- and intermediate-risk groups (*p* = 0.007). As discussed above, the reasons for including some patients of high-risk characteristics were mainly related to the need for treating elderly patients that were referred from hospital of other islands in a highly fragmented territory. The possibility of offering them a safe, one-week treatment favored the inclusion of cases that would have showed some “unfit” tumor characteristics. The ESTRO risk classification confirms its strong predictive role in our series of patients [12].

Multicatheter interstitial brachytherapy spared normal tissues in the breast-reducing late severe toxicity (0% grade 3 in our series, 1% in the GEC ESTRO trial), but it also has the lowest scatter dose possible for other normal tissues, as in the heart and lung [6]. As showed in our series of patients, the 10-year cause-specific survival of 97.7% confirmed the very long survival probability of those patients and the need to follow the as-low-as-reasonably achievable principle [19] for patients with early low-risk breast cancer.

Low-risk breast cancer patients have excellent rates of local control and survival by the combination of multicathether APBI and appropriate systemic treatments (mainly hormonotherapy). Once it is confirmed that radiotherapy volume de-escalation is feasible and safe [11], new proposals arise, suggesting that these low-risk breast cancer patients, especially elderly patients, are good candidates for treatment de-escalation protocols.

In a recently published meta-analysis, including all randomized phase III de-escalating trials [20], suppressing external beam radiation therapy (WBRT) in patients treated with endocrine therapy (ET) was associated with a statistically significant higher rate of in-breast local relapses, but no significant differences were observed in the overall survival [20]. Furthermore, ET is associated with limiting toxicity and a reduced quality of life in patients included in these randomized de-escalation trials [20]. Therefore, randomized trials are already ongoing not only to address the role of hormonal therapy vs. APBI regarding cancer control but also the quality of life [21,22].

The EPOPE trial [21] is a phase III randomized trial comparing standard treatments combining APBI with 5 years of ET versus APBI without ET. This trial is required to accurately analyze the impact of breast cancer adjuvant therapeutic de-escalations in low-risk breast cancers among the elderly. The EUROPA trial [22] is enrolling patients older than 70 years, who were treated by BCS with low risk T1N0 and luminal A tumors. The patients will be randomized to receive either partial-breast irradiation or endocrine therapy. As previously stated, the differences in the patients’ reported outcome measures in terms of QoL seemed to be more relevant for patients in this particular clinical situation than the overall survival. The major endpoints of this study include the following: the QoL assessed by the EORTC QLQ-C30 is required in order to assess the global health status among treatment arms and to also demonstrate a non-inferior local control rate between arms.

## 5. Conclusions

In conclusion, our long-term results confirm that postoperative APBI using multicatheter brachytherapy after BCS is already a standard treatment option for selected patients with early-stage breast cancer. The available results demonstrate the excellent local control rates provided by this technique. Furthermore, the reduced volume irradiated and reduced scattered dose allows for (a) a significant reduction in the dose received by limiting normal tissue organs (heart) and (b) a lack of the breast´s severe late adverse effects.

The limitations of our study include a lack of data on cosmetic results, the inclusion of “unfit” patients that would jeopardize the results obtained when only fit patients were to be included, and the low number of recurrences that would limit the statistical significance of well-established predictive factors in our series of patients.

## Figures and Tables

**Figure 1 cancers-16-01138-f001:**
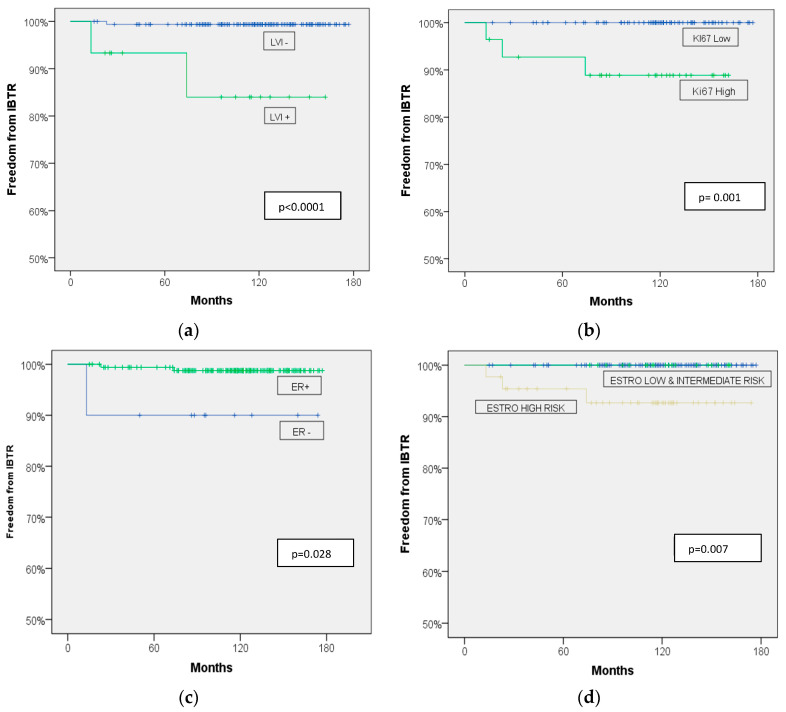
Freedom from IBTR according to (**a**) lymphovascular invasion, (**b**) Ki67 status, (**c**) estrogen receptor status, and (**d**) ESTRO-risk classification.

**Table 1 cancers-16-01138-t001:** Patients’ tumor and postoperative-treatment characteristics.

	Patients	10 y IBTR (95%CI)	*p* Value
Age at diagnosis			
<70 y	113 (62.1%)	0.9% (0–1.8%)	0.267
≥70 y	89 (37.9%)	2.9% (0.7–5.0%)	
Menopausal status			
Premenopausal	11 (6%)	0%	0.652
Postmenopausal	171 (94%)	1.8% (0.7–2.9%)	
Histological type			
Ductal	153 (84.1%)	2.1% (0.9–3.3%)	0.715
Ductal in situ	13 (7.1%)	0%	
Others	16 (8.8%)	0%	
Tumor status			
pTis	13 (7.1%)	0%	0.816
pT1	161 (88.5%)	2% (0.9–3.1%)	
pT2	8 (4.4%)	0%	
Tumor grade			
1	112 (61.5%)	1% (0–2%)	0.411
2	58 (31.9%)	3.5% (1.1–5.9%)	
3	12 (6.6%)	0%	
Lymphovascular Invasion			
Positive	15 (8.2%)	16% (5.4–26.6%)	
Negative	167 (91.8%)	0.6% (0–1.2%)	0.0001
Surgical margins			
≤1 mm	15 (8.2%)	6.7% (0.3–13.1%)	0.10
>1 mm	166 (91.3%)	1.3% (0.4–2.2%)	
Unknown	1 (0.5%)		
ER status			
Positive	172 (94.5%)	1.2% (0.3–2.1%)	0.028
Negative	10 (5.5%)	10% (0.5–19.5%)	
PR status			
Positive	157 (86.3%)	2% (0.9–3.1%)	0.488
Negative	25 (13.7%)	0%	
Her2 status			
Positive	6 (3.3%)	0%	0.736
Negative	157 (86.3%)	2% (0.9–3.1%)	
Unknown	19 (10.4%)		
Ki67 status			
Positive	28 (15.4%)	11.1% (5.0–17.2%)	0.001
Negative	93 (51.1%)	0%	
Unknown	61 (33.5%)		
Molecular subtype			
Luminal	174 (95.6%)	1.8% (0.8–2.8%)	0.710
Non luminal	8 (4.4%)	0%	
ASTRO group			
Eligible	86 (47.3%)	0%	0.227
Cautionary	72 (39.6%)	2.9% (0.9–4.9%)	
Ineligible	24 (13.1%)	4.3% (0–8.6%)	
ESTRO risk			
Low	118 (64.8%)	0%	0.007
Intermediate	20 (11.0%)	0%	
High	44 (24.2%)	7.3% (3.2–11.4%)	

**Table 2 cancers-16-01138-t002:** Patients’ relapse and survival rates at different time points.

	3 Years	5 Years	8 Years	10 Years	13 Years
	Patients at Risk (175)	Patients at Risk (166)	Patients at Risk (136)	Patients at Risk (88)	Patients at Risk (22)
IBTR	1.1% (95%CI: 0.3–1.9%)	1.1% (95%CI: 0.3–1.9%)	1.7% (95%CI: 0.7–2.7%)	1.7% (95%CI: 0.7–2.7%)	1.7% (95%CI: 0.7–2.7%)
Regional Failure	0.6% (95%CI: 0.0–1.2%)	0.6% (95%CI: 0.0–1.2%)	0.6% (95%CI: 0.0–1.2%)	0.6% (95%CI: 0.0–1.2%)	0.6% (95%CI: 0.0–1.2%)
Distant Failure	1.7% (95%CI: 0.8–2.6)	1.7% (95%CI: 0.8–2.6)	2.3% (95%CI: 1.2–3.4)	2.3% (95%CI: 1.2–3.4)	5.2% (95%CI: 2.1–8.3)
Disease-Free Survival	97.2% (95%CI: 96–98.4)	96.7% (95%CI: 95.4–98)	94.9% (95%CI: 93.2–96.6)	94.9% (95%CI: 93.2–96.6)	91.6% (95%CI: 88–95.2)
Cause-Specific Survival	98.3% (95%CI: 97.3–99.3)	98.3% (95%CI: 97.3 99.3)	97.7% (95%CI: 96.6–98.8)	97.7% (95%CI: 96.6–98.8)	97.7% (95%CI: 96.6–98.8)
Overall Survival	96.7% (95%CI: 95.4–98)	93.3% (95%CI: 91.4–95.2)	87.0% (95%CI: 84.5–89.5)	80.7% (95%CI: 77.5–93.9)	73.7% (95%CI: 69.1–78.3)

## Data Availability

The data presented in this study are available on request from the corresponding author.

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
