# Peer review of "Ten-Year Results of Accelerated Partial-Breast Irradiation with Interstitial Multicatheter Brachytherapy after Breast-Conserving Surgery for Low-Risk Early Breast Cancer"

_cancers, 2024, doi:10.3390/cancers16061138_

Round 1

Reviewer 1 Report

Comments and Suggestions for Authors

The present manuscript is an update (10 years follow up)of a retrospective analysis published earlier by these authors. The data are well presented and useful; illustrations are appropriate. The results are in line with current follow up of breast cancer patients, testifying to successful treatment. 

Major comment: The authors refer to  ref [17] for “our” previous methods and results. Is this correct? The refereed publication is by an Italian team, namely, Takanen S, Gambirasio A, Gritti G et al Breast cancer electron intraoperative radiotherapy: assessment of preoperative selection factors from a retrospective analysis of 758 patients and review of literature. Breast Cancer Res Treat. 2017 Sep;165(2):261-271. Probably meant is Nieves G. Rodríguez-Ibarria et al. Breast 2020; 52/45-49. 

Details: 

All abbreviations need explanation, IBRT or IBTR? 

Use decimal points (0.6%) instead of commas (0,6 %).  

Avoid redundancy: Numerical values in the Table need not be repeated in the text. 

Lay out: Why does Table 2 precede Table 1?

Author Response

Response to reviewer 1

1.-Major comment: The authors refer to  ref [17] for “our” previous methods and results. Is this correct? The refereed publication is by an Italian team, namely, Takanen S, Gambirasio A, Gritti G et al Breast cancer electron intraoperative radiotherapy: assessment of preoperative selection factors from a retrospective analysis of 758 patients and review of literature. Breast Cancer Res Treat. 2017 Sep;165(2):261-271. Probably meant is Nieves G. Rodríguez-Ibarria et al. Breast 2020; 52/45-49.

Response: we corrected this incorrect citation, Now reads

[17]Rodriguez-Ibarria NG, Pinar MB, García L, Cabezón MA, Lloret M, Rey-Baltar MD, Rdguez-Melcón JI, Lara PC.Accelerated partial breast irradiation with interstitial multicatheter brachytherapy after breast-conserving surgery for low-risk early breast cancer. Breast. 2020 Aug;52:45-49. doi: 10.1016/j.breast.2020.04.008. Epub 2020 Apr 22.

[18]Takanen S, Gambirasio A, Gritti G et al Breast cancer electron intraoperative radiotherapy: assessment of preoperative selection factors from a retrospective analysis of 758 patients and review of literature. Breast Cancer Res Treat. 2017 Sep;165(2):261-271

Details:

2.-All abbreviations need explanation, IBRT or IBTR?

Response: IBRT abbreviation has been explained and uniformly corrected through the paper. Now reads:

The 10-y ipsilateral breast tumor recurrence (IBTR) was 1.7% (CI95%: 0.7-2.7%) and the cause specific survival was 94.9% (95% CI 93.2-96.6%).

3.-Use decimal points (0.6%) instead of commas (0,6 %). 

Response: following reviewer´s indications, we used decimal points instead of commas through the paper

4.-Avoid redundancy: Numerical values in the Table need not be repeated in the text.

Response: following reviewer´s indications, the numerical values in the text were as reduced as possible in order to avoid redundancy.

5.-Lay out: Why does Table 2 precede Table 1?

Response: Table 1 is cited before Table 2. The final paper lay-out do not rely in our decision

Reviewer 2 Report

Comments and Suggestions for Authors

The article titled " Ten years results of accelerated partial breast irradiation with interstitial multicatheter brachytherapy after breast-conserving surgery for low-risk early breast cancer." by Nieves Gloria Rodríguez-Ibarria et al. provides valuable insights. However, there are several issues that need to be addressed:

1. The study had a retrospective design, which might introduce selection bias and limit the generalizability of the findings.

2. The sample size was relatively small, which could limit the statistical analysis and precision of the results.

3. The follow-up period of 10 years might not be sufficient to assess the long-term safety and efficacy of the treatment, as some adverse effects could be emerged later.

4. The study only included low-risk breast cancer patients, and the results might not apply to other risk groups.

5. Authors did not provide detailed information on the characteristics of the patients included, such as tumor size, lymph node involvement, hormone receptor status, and HER2/neu status.

6. Author did not compare the efficacy and safety of multicatheter brachytherapy with other treatment options, such as whole breast irradiation or mastectomy.

7. Authors did not provide info on the explicit techniques used for multicatheter brachytherapy, such as dose prescription, fractionation scheme, and dose constraints

8. The authors did not report on patient-reported outcomes, such as quality of life or cosmetic results, which are important considerations for breast cancer patients.

9. The study lacks data on adverse events related to multicatheter brachytherapy, such as infection, fibrosis, or cosmetic changes

10. Authors did not contemplate the long-term oncological results, such as local recurrence rates, distant metastasis rates, or overall survival.

Comments on the Quality of English Language

Moderate editing of English language required

Author Response

  1. The study had a retrospective design, which might introduce selection bias and limit the generalizability of the findings.

Response: patients were included in the study at the time of APBI and followed prospectively under the Spanish RD1566/1998 regulation for Radiation Therapy Quality Assurance. That regulation makes follow-up compulsory in terms of toxicity scoring and oncological outcomes.

  1. The sample size was relatively small, which could limit the statistical analysis and precision of the results.

Response: We agree that the sample size could be larger, but our study has more patients than 13 of the 20 studies published until 2022,

[ Cozzi S, Augugliaro M, Ciammella P, Botti A, Trojani V, Najafi M, Blandino G, Ruggieri MP, Giaccherini L, Alì E, Iori F, Sardaro A, Finocchi Ghersi S, Deantonio L, Gutierrez Miguelez C, Iotti C, Bardoscia L.The Role of Interstitial Brachytherapy for Breast Cancer Treatment: An Overview of Indications, Applications, and Technical Notes.Cancers (Basel). 2022 May 23;14(10):2564. doi: 10.3390/cancers14102564].

  1. The follow-up period of 10 years might not be sufficient to assess the long-term safety and efficacy of the treatment, as some adverse effects could be emerged later.

Response: We agree that follow-up could be longer, but a mean follow-up of 10 years seemed to be enough for publications in the Lancet Oncology journal of the GEC-ESTRO trial. [Strnad V, Polgár C, Ott OJ, et al Accelerated partial breast irradiation using sole interstitial multicatheter brachytherapy compared with whole-breast irradiation with boost for early breast cancer: 10-year results of a GEC-ESTRO randomised, phase 3, non-inferiority trial. Lancet Oncol. 2023 Mar;24(3):262-272]

In that article the authors concluded: “Interpretation: Postoperative APBI using multicatheter brachytherapy after breast-conserving surgery in patients with early breast cancer is a valuable alternative to whole-breast irradiation in terms of treatment efficacy and is associated with fewer late side-effects”

Therefore, 10 years follow-up seemed to be enough for authors and Lancet Oncology Editors to assess efficacy and safety in this particular clinical situation.

  1. The study only included low-risk breast cancer patients, and the results might not apply to other risk groups.

Response: We agree that our patient´s low-risk breast cancer characteristics, are only a group, among breast cancer patients treated with postoperative radiotherapy. Fortunately, the international scientific societies guidelines already cited in the article [12-14], clearly describe the risk classifications for the suitable patients for this treatment approach.

Furthermore, we have included patients classified as

  1. a) suitable, cautionary and unsuitable for ASTRO guidelines and
  2. b) low, intermediate and high risk for ESTRO guidelines.

Therefore, although all patients were low risk cases, considering tumor extension, we included patients with different risk situations for APBI.

  1. Authors did not provide detailed information on the characteristics of the patients included, such as tumor size, lymph node involvement, hormone receptor status, and HER2/neu status.

Response: We do not understand the reviewer´s comment. In order to avoid redundancy we wrote:

Patient´s characteristics were previously described [17] (Table 1). The mean age at treatment was 67 years (range 50–92) and most patients showed a luminal molecular subtype cancer (95.6%) Most patients were considered cautionary or ineligible by ASTRO guidelines but low risk for GEC-ESTRO characteristics.

We refer the reviewer to Table 1 where all the patient´s and tumor characteristics are described.

Table 1.Patient´s, tumor and postoperative treatment characteristics.

Patients

10 y IBTR(95%CI)

P value

Age at diagnosis

<70 y

113 (62.1%)

0.9%(0-1.8%)

0.267

³ 70y

89 (37.9%)

2.9%(0.7-5.0%)

Menopausal status

Premenopausal

11 (6%)

0%

0.652

Postmenopausal

171 (94%)

1.8% (0.7-2.9%)

Histological type

Ductal

153 (84.1%)

2.1%(0.9-3.3%)

0.715

Ductal in situ

13 (7.1%)

0%

Others

16 (8.8%)

0%

Tumour status

pTis

13 (7.1%)

0%

0.816

pT1

161 (88.5%)

2%(0.9-3.1%)

pT2

8 (4.4%)

0%

Tumour grade

1

112 (61.5%)

1%(0-2%)

0.411

2

58 (31.9%)

3,5%(1.1-5.9%)

3

12 (6.6%)

0%

Lymphovascular Invasion

Positive

15

16%(5.4-26.6%)

Negative

167

0,6%(0-1.2%)

0.0001

Surgical margins

<=1mm

15 (8.2%)

6.7%(0.3-13.1%)

0.10

>1mm

166 (91.3%)

1.3%(0.4-2.2%)

Unknown

1 (0,5%)

ER status

Positive

172 (94.5%)

1.2%(0.3-2.1%)

0.028

Negative

10 (5.5%)

10%(0.5-19.5%)

PR status

Positive

157 (86.3%)

2%(0.9-3.1%)

0.488

Negative

25 (13.7%)

0%

Her2 status

Positive

6 (3.3%)

0%

0.736

Negative

157 (86.3%)

2%(0.9-3.1%)

Unknown

19 (10.4%)

Ki67 status

Positive

28 (15,4%)

0%

0.001

Negative

93 (51,1%)

11.1%(5.0-17.2%)

Unknown

61 (33,5%)

Molecular subtype

Luminal

174 (95.6%)

1.8%(0.8-2.8%)

0.710

Non luminal

8 (4.4%)

0%

ASTRO group

Eligible

86 (47.3%)

0%

0.227

Cautionary

72 (39.6%)

2.9%(0.9-4.9%)

Ineligible

24 (13.1%)

4.3%(0-8.6%)

ESTRO risk

Low

118 (64.8%)

0%

0.007

Intermediate

20 (11.0%)

0%

High

44 (24.2%)

7.3%(3.2-11.4%)

  1. Author did not compare the efficacy and safety of multicatheter brachytherapy with other treatment options, such as whole breast irradiation or mastectomy.

Response: At present time, there is no role for mastectomy in these early low-risk, non-multifocal, node negative patients. Therefore, it is not possible to find modern publications of mastectomy in this clinical situation.

Regarding the comparison with WBI, we already cited the mature Phase III trial, that showed that APBI had non-inferior efficacy with lower side effects compared to WBI combined with tumor bed boost.

We included a paragraph in the Discussion section to compare our results with those of the Phase III trial

“Results of our treatment protocol related to regional relapse (0.6%) and contralateral breast relapse (1.65%) are in good accordance with the GEC ESTRO results [11]. Data on distant disease-free survival (94.9%) and cause specific survival (97.7%) are slightly better than the GEC-ESTRO study (Disease Free Survival at 10y: 84,89%). The favorable characteristics of our series of patients, support these excellent results. Interestingly the overall survival is lower in our series (80.7% at 10 y) compared to those observed in the reference phase III trial [11]. The advanced age at diagnosis of the included patients (40% of the patients were older than 70 years at diagnosis) would explain this low survival in the situation of excellent cancer control figures.”

  1. Authors did not provide info on the explicit techniques used for multicatheter brachytherapy, such as dose prescription, fractionation scheme, and dose constraints.

Response: We agree with the reviewer´s indications. Now reads:

The treatment protocol has been previously reported [17]. In short, all patients had interstitial multiplanar implants to effectively cover the clinical target volume (CTV) including both the surgical cavity and a safety margin (at least 20mm). All patients had a pre-planning computed tomography (CT) and planning CT scans (for treatment planning and documentation of multicatheter brachytherapy). A standard 1.5cm catheter spacing was used. Dose prescription and calculation were in agreement, with the International Commission of Radiation Units and Measurements (ICRU) report 58.

Dose-volume histogram analysis was performed to evaluate dose coverage (100% of the dose covered at least 90% of CTV), homogeneity (non-uniformity ratio below 0.35), and normal tissue limits (skin dose below 70% of the prescribed dose). Accelerated partial breast irradiation was delivered, with high-dose-rate (HDR) multicatheter brachytherapy, in 8 fractions of 4 Gy, always separated at least 6 hours each other. The treatment was administered in 5 consecutive days to a total dose of 32 Gy.

  1. The authors did not report on patient-reported outcomes, such as quality of life or cosmetic results, which are important considerations for breast cancer patients.

Response: we agree with the reviewer that PRO, and cosmetic results are important. These important considerations were not included as primary or secondary objectives of the study.

  1. The study lacks data on adverse events related to multicatheter brachytherapy, such as infection, fibrosis, or cosmetic changes.

Response: Severe (grade >=3) acute and late toxicity was considered a secondary endpoint in the study [17]. After 10 years mean follow-up severe acute toxicity cannot change from the first analysis already published and new severe late toxicity does not appear.

Treatment related acute and late toxicity rates were already reported [17]. After a mean follow-up more than 10 years, no severe grade 3–4 late adverse effects were recorded.

  1. Authors did not contemplate the long-term oncological results, such as local recurrence rates, distant metastasis rates, or overall survival.

Response: we do not understand the reviewer´s comments. We described in the text the most relevant data on local recurrence, distant metastases, and survival:

Follow-up was closed on October 15th,2023. The mean follow-up for survivors was 123.27+/-29 months (median123 months, range 17–177) Four patients were lost of follow-up at 17, 42, 46 and 51 months after treatment, as they moved on to another Spanish region.

Three out of 182 patients have developed local recurrence (LR) during follow-up, two relapses in the tumor bed (23 and 74 months) and the other elsewhere in the ipsi-lateral breast at 14 months. The IBTR risk at 5 years was 1.1% (95% CI, 0.3–1.9) and 1.7% (CI95%: 0.7-2.7%) at 10 years.

One case of isolated regional recurrence was reported (5 &10-year cumulative incidence rate 0.6% (95% CI, 0.0–1.2%). Two other cases showed axillary relapse with concomitant systemic relapse at 76 and 153 months respectively. Five patients showed dis-tant relapse at 14, 22, 24, 76, and 153 months, respectively. The cumulative incidence of distant metastases at 5 years and 10 years were 1.7% (95% CI, 0.8–2.6) and 2.3% (95% CI, 1.2–3.4) respectively (Table 2).

During follow-up three patients developed contralateral breast cancer (1.65%) and thirteen patients developed second primary cancer (7.14%). The 5 & 10-year disease-free survival rates were 96.7% (95% CI, 95.4–98) and 94,9 (95% CI, 93.2–96.6) respective-ly. Thirty -six patients died during follow-up, four of them due to breast cancer and 32 patients by other non-breast cancer disease. The 5&10year cumulative cause-specific survival were 98.3% (95% CI,97.3-99.3) and 97.7% (95% CI, 96.6–98.8) respectively (Ta-ble 2). The overall survival at 5 &10 year was 93.3±1.9% and 80.7±3.2%, respectively.

Furthermore, in Table 2, data on detailed oncological outcomes at 3,5,8,10 and 12 years are shown.

Table 2.- Patient´s relapse and survival rates at different time points

3 years

5 years

8 years

10 years

13 years

Patients at risk

(175)

Patients at risk

(166)

Patients at risk

(136)

Patients at risk

(88)

Patients at risk

(22)

 IBTR

1.1%

(CI95%: 0.3-1.9%)

1.1%

(CI95%: 0.3-1.9%)

1.7%

(CI95%: 0.7-2.7%)

1.7%

(CI95%: 0.7-2.7%)

1.7%

(CI95%: 0.7-2.7%)

Regional Failure

0,6%

(CI95%: 0.0-1.2%)

 0,6%

(CI95%: 0.0-1.2%)

0,6%

(CI95%: 0.0-1.2%)

0,6%

(CI95%: 0.0-1.2%)

0,6%

(CI95%: 0.0-1.2%)

Distant Failure

1.7%

(95% CI, 0.8–2.6)

1.7%

(95% CI, 0.8–2.6)

2,3%

(95% CI, 1.2–3.4)

2,3%

(95% CI, 1.2–3.4)

   5,2%      

(95% CI, 2.1–8.3)

Disease Free Survival

97,2%

(95% CI, 96–98,4)      

96.7%

(95% CI, 95.4–98)

 94,9%

(95% CI, 93.2-96.6)

 94,9%

(95% CI, 93.2–96.6)

91,6%        ,036

(95% CI, 88–95.2)

Cause Specific Survival

98.3%

(95% CI,97.3-99.3)

98.3%

(95% CI,97.3 99.3)

 97,7%

(95% CI, 96.6–98.8)

97,7%

(95% CI, 96.6–98.8)

97,7%

(95% CI, 96.6–98.8)

Overall Survival

96,7%      

(95% CI,95.4-98)

93,3%       

(95% CI,91.4-95.2)

87,0%

(95% CI,84.5-89.5)

80,7%      

(95% CI,77.5-93.9)

73,7%      

(95% CI,69.1-78.3)

Round 2

Reviewer 2 Report

Comments and Suggestions for Authors

Accept in present form

Comments on the Quality of English Language

Moderate editing of English language required